# Genetic Diversity of *Plasmodium vivax Surface Ookinete Protein Pvs25* and Host Genes in Individuals Living along the Thai–Myanmar Border and Their Relationships with Parasite Density

**Abdifatah Abdullahi Jalei [1]** 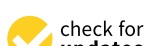**, Wanna Chaijaroenkul [1,2] and Kesara Na-Bangchang [1,2,3,*]**

[1] Graduate Studies, Chulabhorn International College of Medicine, Thammasat University Rangsit Campus, 99 Moo 18 Phaholyothin Road, Klong Luang District, Pathumthani 12121, Thailand; abdifatah.abd@dome.tu.ac.th (A.A.J.); cwanna@tu.ac.th (W.C.)

[2] Center of Excellence in Pharmacology and Molecular Biology of Malaria and Cholangiocarcinoma, Chulabhorn International College of Medicine, Rangsit Campus, Thammasat University, 99 Moo 18 Phaholyothin Road, Klong Luang District, Pathum Thani 12121, Thailand

[3] Drug Discovery and Development Center, Thammasat University, Rangsit Campus, 99 Moo 18 Phaholyothin Road, Klong Luang District, Pathum Thani 12121, Thailand

[*] Correspondence: kesaratmu@yahoo.com or nkesara@tu.ac.th; Tel.: +662-564-4440-79 (ext. 1803); Fax: +662-564-4398

**Abstract:** *Plasmodium vivax* (*Pv*) accounts for over 50% of malaria cases in Latin America and Asia. Despite a significant reduction in *Pv* transmission in Thailand, the parasite remains endemic to the border areas. This study aimed to investigate the genetic diversity of the parasites and the host factors, as well as their relation to parasite density in *Pv* isolates, along the Thai–Myanmar border. Genetic variations in *Pv* markers, specifically the ookinete surface protein *Pvs25*, and host genes, including Toll-like receptor 6 (TLR6), TLR9, TIR Domain-containing adaptor protein (TIRAP), Toll-interacting protein (TOLLIP), Duffy antigen receptor for chemokines (DARC), and intercellular adhesion molecule 1 (ICAM-1), were investigated using polymerase chain reaction (PCR) with restriction fragment length polymorphism (RFLP). A total of 548 PCR-positive *Pv* samples collected from Tak and Kanchanaburi provinces during two periods (2006–2007 and 2014–2016) were included in the study. *Pvs25* exhibited four haplotypes, with H1 (EGTKV) being the most prevalent in both provinces. Kanchanaburi isolates exhibited greater genetic diversity than Tak isolates. No significant deviations from neutrality were observed for *Pvs25* in either area. ICAM-1 and TOLLIP s3750920 heterozygous carriers had greater median parasite densities than homozygous mutants. The TLR9 rs187084 T genotype had a significantly higher parasite density than the non-T genotype. The findings underscore the significant association between the rs3750920 C/T, rs5498 A/G, and rs187084 T genotypes and high parasite density in patients infected with *Pv*, highlighting their potentially critical role in malaria susceptibility.

**Keywords:** genetic diversity; *Plasmodium vivax*; *Pvs25*; TIRAP; TLR; TOLLIP; ICAM-1; DARC; Thai–Myanmar

## 1. Introduction

Malaria continues to pose a significant public health challenge in endemic countries despite tremendous achievements in its control in recent years [1]. The emergence of *Plasmodium falciparum* (*Pf*) and *Plasmodium vivax* (*Pv*) strains resistant to chloroquine and sulfadoxine–pyrimethamine underscores the need for new malaria control strategies such as vaccines [2]. Although several *Plasmodium* vaccine candidates have been intensively studied, little progress has been made beyond preclinical evaluation. The circumsporozoite protein (CSP), *Pv* ookinete surface protein *Pvs25*, and Duffy-binding protein

(PvDBP_RII) [3,4] trigger strong antibody responses capable of inhibiting infection. However, none has achieved complete protection against malaria. In malaria-endemic regions, the disease is believed to be the driving force for the natural selection of genetic variations in erythrocyte proteins, hemoglobin, and other immune effectors [5]. Malaria parasites are initially recognized by components of the host's innate response, facilitated by pattern recognition receptors (PRRs), which play essential roles in recognizing pathogen-derived components and initiating innate immune signaling pathways. Toll-like receptors (TLRs) are among the most important PRRs crucial in inducing an immune response to malaria [6]. The immune response to *Pv* is not as well understood as that of the more extensively studied *Pf* [7]. The *Pv* genome is enriched in CpG motifs, which can bind TLR9, triggering type I interferon (IFN) production [8]. By invading reticulocytes expressing class I MHC, *Pv* may provide an alternative pathogen control route via cytolysis by antigen-specific CD8+ T cells [9]. However, limited in vivo information exists on the number and function of activated T cells, and field studies have been conducted to compare host responses to *Pv* and *Pf*. Challenges in immune response to *Pv* in endemic settings include differences in host age, parasite load, parasite genotype, and exposure history [10].

Host genetic influences have been shown to explain approximately one-third of the diversity in the risk of severe or complicated malaria, with numerous genetic variations associated with malaria protection [11]. TLRs and their signaling pathways are crucial for triggering immune responses in malaria; however, determining the specific receptors contributing to immunopathogenesis is essential [6]. Studies exploring the in vivo relevance of TLRs in cytokine responses and host susceptibility to *Pv* infection have indicated an association between TLR1 I602S variants and mild clinical manifestations, TLR6 S249P homozygotes and asymptomatic malaria, and TLR9 1486C/T variants and high parasitemia [12]. Toll-interacting protein (TOLLIP), a negative regulator of the human TLR signaling cascade, influences interleukin-6 (IL6), tissue necrosis factor-alpha (TNF-$\alpha$), and IL10 expression [13], showing that individuals homozygous for the TOLLIP rs3750920 T allele have twice the risk of developing malaria compared to those homozygous for the C allele [14]. The TOLLIP rs5743899 T allele has been associated with susceptibility to *Pv* infection in the Amazonas state of Brazil [14]. Other studies in Brazil have also confirmed an association between TLR9 polymorphisms, disease susceptibility, and parasitemia control in individuals infected with *Pv* [15,16]. Identifying the genes involved in susceptibility or resistance to *Pv* and *Pf* may pave the way for a better understanding of the disease and the development of new strategies to treat or prevent the disease in individuals with specific genotypes. Although genetic association studies remain controversial [17], obtaining additional data from different malaria-endemic regions is essential for understanding the role of TLRs and adapter proteins in *Pv* susceptibility or severity. The present study investigated the genetic diversity of the *Pv* marker (*Pvs25*) and polymorphisms in human host genes (TLR6, TLR9, TIR Domain-containing adaptor protein (TIRAP), TOLLIP, Duffy antigen receptor for chemokines (DARC), and intercellular adhesion molecule 1 (ICAM-1)) and their relation to parasite density in *Pv* isolates from areas along the Thai–Myanmar border.

## 2. Materials and Methods

### 2.1. Ethical Approval

The study received approval from the Thammasat University Ethics Committee for Research in Human Subjects, Thailand (No. 082/2560). Informed consent was obtained from all participants before specimen collection.

### 2.2. Study Area and Sample Processing

Dried blood spot (DBS) samples were collected over two periods (2006–2007 and 2014–2016) from 548 symptomatic patients living in Mae Sot district (Tak province) and Sai Yok district (Kanchanaburi province), two districts along the Thai–Myanmar border area (Figure 1). Patients were diagnosed with *Pv* infection through microscopic examination of Giemsa-stained blood smears under a light microscope (1000× magnification), and

the diagnosis was confirmed by nested PCR amplification. Asexual-stage parasites were counted three times against 200 white blood cells (WBCs) in a thin blood smear, and the parasite density (per microliter) was calculated. Genomic DNA for genotyping was prepared using a QIAmp DNA Kit (Qiagen, Chatsworth, CA, USA).

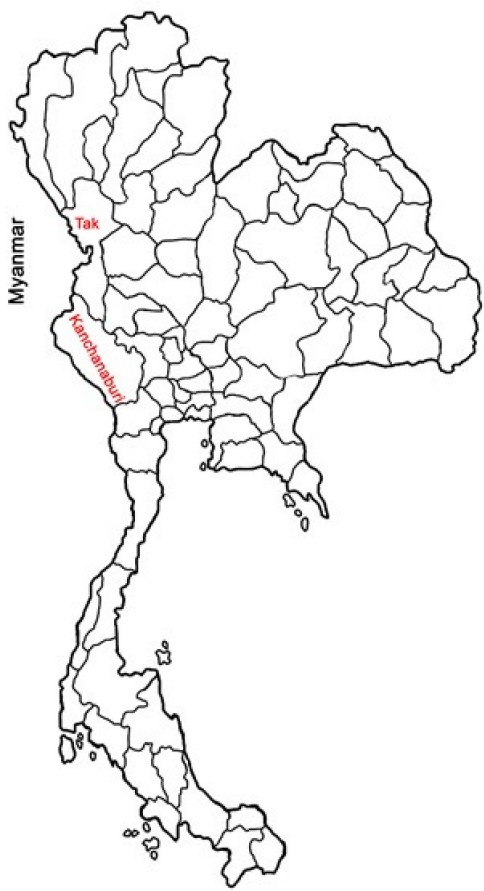

**Figure 1.** Map of Thailand: the sample collection areas are along the Thai–Myanmar border areas of Tak province and Kanchanaburi province.

### 2.3. Polymorphism Host Genotyping

The polymorphisms of the following genes were investigated using polymerase chain reaction-restriction fragment length polymorphism (PCR-RFLP): DARC, ICAM-1, TOL-LIP (rs3750920 and rs5743899), TIRAP S180L (rs8177374), TLR6 S249P (rs5743810), TLR9 1237C/T (rs187084), and TLR9 1486C/T (rs5743836). The products were incubated with 5 U of their respective restriction enzymes overnight and, subsequently, separated via electrophoresis on a 3% agarose gel.

### 2.4. Statistical Analysis

The statistical analysis was performed using SPSS (IBM Corp., Armonk, NY, USA). Genotype and allele frequencies were calculated for host genotype polymorphisms, and the Hardy-Weinberg equilibrium (HWE) for each single nucleotide polymorphism (SNP) was determined using the Gene-Calc tool (https://gene-calc.pl/hardy-weinberg-page, accessed on 23 February 2023). The Kruskal-Wallis test with Dunn's post hoc test or Mann-Whitney U test was performed to determine the association between genotypes, gene alleles, and *Pv* parasite density. The statistical significance level was set at $p < 0.05$.

### 2.5. Sequencing and Bioinformatic Analysis of Pvs25 Sequences

The SNPs in *Pvs25* were investigated using nested PCR [18], and the amplified products were purified using a PCR purification kit (Qiagen). Sequencing was conducted in

both directions using an automated ABI 3730xl DNA sequencer (Macrogen Inc., Seoul, Republic of Korea). The obtained DNA sequences were manually trimmed, edited, and aligned with the reference strain *Pvs25* Sal-I (AF083502), using the ClustalW alignment tool in BioEdit software version 7.2 (Carlsbad, CA, USA). Haplotypes were identified using ALTER software (https://www.sing-group.org/ALTER/, accessed on 20 March 2023). The chi-square test was used to evaluate differences in amino acid substitutions between the two periods. Furthermore, genetic diversity indices, including segregating polymorphic sites (S), average pairwise nucleotide differences (K), haplotype diversity (Hd), and nucleotide diversity ($\pi$), were computed using DnaSP version 5.10.00 software (DNASTAR, Madison, WI, USA).

### 2.6. Construction of the Global Database of Pvs25

To compare the *Pvs25* haplotype diversity of *Pv* in Thailand with global sequence data, a search was conducted in the PubMed and Google Scholar databases for articles reporting *Pvs25* sequences. Accession numbers of the reported sequences were searched in the NCBI public databases and downloaded in FAST format. The comparative analysis involved comparing the obtained *Pvs25* sequences with those originating from diverse geographical regions, including Thailand [19], Myanmar [20], China [21,22], South Korea [23,24], India [25], Iran [26], North Korea, Bangladesh, Indonesia, Papua New Guinea (PNG), Mauritania, Venezuela, Brazil, Colombia [27], and Mexico [28].

## 3. Results

### 3.1. Study Population and Demographic Data

A total of 548 PCR-positive *Pv* samples collected from the Thai–Myanmar border over two periods (2006–2007 and 2014–2016) were included in the analysis. These samples comprised 208 and 340 *Pv*-infected blood samples from Kanchanaburi and Tak provinces, respectively (69.3% from male patients and 30.7% from female patients). The parasite density for males (69.3% of all cases) ranged from 2,020 to 260,400 parasites/µL, while that for females (30.7% of all cases) ranged from 3,020 to 203,280 parasites/µL. The successful amplification rates for the studied host genes were as follows: DARC (88.7%), ICAM-1 (88.0%), TOLLIP rs3750920 (91%), TOLLIP rs5743899 (92%), TIRAP (89%), TLR6 rs5743810 (86.7%), TRL9 rs5743836 (88.5%), and TLR9 rs187084 (88.9%).

### 3.2. Genotype and Allele Distribution of the Host Genes

The genotypes and allele frequencies of immune-related genes (TIRAP, TLR6, TLR 9, and TOLLIP) and malaria parasite-binding genes (ICAM-1 and DARC) are shown in Table 1. Among the immune-related genes, the TIRAP S180L genotype frequencies were CC (88.9%), CT (7.1%), and TT (4.0%). The HWE analysis indicated a significant deviation from the expected frequencies ($p < 0.001$). The TLR6 rs5743810 genotype frequencies were TT (92.6%), TC (6.3%), and CC (1%), with significant deviations from the expected frequencies according to the HWE analysis ($p < 0.001$). For TLR9 rs187084, the genotype distribution was 36.9% in TT, 43.4% in CT, and 19.7% in CC, with the HWE analysis showing deviation ($p = 0.024$). However, for TLR9 rs5743836, the genotype distribution was 2.9% in TT and 97.1% in CC. In the TOLLIP gene, two SNPs (rs3750920 and rs5743899) were examined; rs3750920 exhibited a genotype distribution of CC in 36.1%, CT in 48.7%, and TT in 15.2%. The HWE analysis revealed no significant deviation from the expected frequencies ($p = 0.685$). For rs5743899, the genotype distribution was AA in 41.7%, AG in 49.6%, and GG in 8.7%, with the HWE analysis indicating a significant deviation ($p = 0.011$). The ICAM-1 gene analysis found a genotype distribution of AA in 59.8%, AG in 34.2%, and GG in 6.0%. The HWE analysis revealed no significant deviation ($p = 0.411$). The DARC genotype frequencies were FYA (81.5%), FYB (15.0%), and FYAB (3.5%).

**Table 1.** The genotype and allele distributions of host genes associated with *Pv* parasite density.

| Gene Type | Gene | No. Successfully Amplified | SNP | Homozygous Wild Type | Heterozygous Genotype | Homozygous Mutant | Total Alleles, *n* (%) | | HWE, *p*-Value |
|---|---|---|---|---|---|---|---|---|---|
| | | | | Genotypes, *n* (%) | | | | | |
| Immune-related genes | TIRAP | 488 | rs8177374 | CC: 448 (88.9) | CT: 36 (7.1) | TT: 20 (4.0) | C: 932 (92.5) | T: 76 (7.5) | <0.001 |
| | TLR6 | 475 | rs5743810 | TT: 440 (92.6) | CT: 30 (6.3) | CC: 5 (1) | T: 910 (95.8) | C: 40 (4.2) | <0.001 |
| | TLR9 | 487 | rs187084 | TT: 180 (36.9) | CT: 212 (43.4) | CC: 96 (19.7) | T: 572 (58.7) | C: 402 (41.3) | 0.024 |
| | | 485 | rs5743836 | TT: 14 (2.9) | CT: 0 (0.0) | CC: 471(97.1) | T: 28 (2.9) | C: 942 (97.1) | <0.001 |
| | TOLLIP | 499 | rs3750920 | CC: 180 (36.1) | CT: 243 (48.7) | TT: 76 (15.2) | T: 603 (60.4) | C: 395 (39.6) | 0.685 |
| | | 504 | rs5743899 | AA: 210 (41.7) | AG: 250 (49.6) | GG: 44 (8.7) | A: 670 (66.5) | G: 338 (33.5) | 0.011 |
| Cytoadherence genes | ICAM-1 | 482 | rs5498 | AA: 288 (59.8) | AG: 165 (34.2) | GG: 29 (6.0) | A: 741(76.9) | G: 223 (23.1) | 0.411 |
| | DARC | 486 | - | FYA/FYA: 396 (81.5) | FYA/FYB: 17 (3.5) | FYB/FYB: 73 (15.0) | FYA: 809 (83.2) | FYB: 163 (16.8) | <0.001 |

The data are presented as numbers (*n*) and percentages (%). HWE, Hardy-Weinberg equilibrium. Statistically significant at *p*-value < 0.05. SNP, single nucleotide polymorphism.

### 3.3. Associations between Host Genotype Variants and Parasite Density

The influence of host genetic polymorphisms on parasitemia was evaluated by comparing the median parasite densities (parasites/µL of blood) in carriers of each genotype (Table 2). Significant associations were observed for ICAM-1, TLR9 rs187084, and TOLLIP rs3750920. Specifically, individuals with the ICAM-1 A/G genotype had greater median parasite density than GG carriers (4.34 vs. 4.10 patients/µL) in a dominant inheritance model ($p = 0.02$). Patients with the rs187084 T/T genotype had significantly greater parasitemia than those with non-TT genotype (4.34 vs. 4.15 parasites/µL, $p = 0.007$), with Bonferroni-adjusted *p*-values of 0.014 (TT vs. CT) and 0.043 (TT vs. CC). No significant differences were observed for the remaining genotypes.

**Table 2.** Analysis of the associations between the studied genotypes and *Pv* parasite density.

| Polymorphism | Genotype | Log10 Parasite Density | *p*-Value (*p*-corrected) |
|---|---|---|---|
| TIRAP (S180L) | CC | 4.28 (2.17–5.42) | 0.303 |
| | CT | 4.15 (3.34–5.26) | |
| | TT | 4.15 (3.59–5.26) | |
| TLR6 (rs5743810) | TT | 3.95 (3.65–4.65) | 0.626 |
| | CT | 4.34 (3.61–5.26) | |
| | CC | 4.19 (2.71–5.42) | |
| TLR9 (rs5743836) | TT | 4.14 (3.48–4.91) | 0.501 |
| | CC | 4.22 (2.71–5.42) | |
| TLR9 (rs187084) | TT | 4.34 (3.31–5.42) | 0.007 (0.014) [1] |
| | CT | 4.15 (3.15–5.26) | 0.007 (0.043) [2] |
| | CC | 4.17 (3.15–4.77) | |
| TOLLIP (rs3750920) | CC | 4.17 (3.31–5.26) | 0.039 (0.033) [3] |
| | CT | 4.28 (3.44–5.42) | |
| | TT | 4.15 (2.71–4.66) | |
| TOLLIP (rs5743899) | AA | 4.26 (3.09–5.42) | 0.182 |
| | AG | 4.27 (2.71–5.31) | |
| | GG | 4.11 (3.34–5.08) | |
| ICAM-1 (rs5498) | AA | 4.15 (3.15–5.42) | 0.012 (0.020) [4] |
| | AG | 4.34 (3.44–5.31) | |
| | GG | 4.10 (2.71–4.68) | |
| DARC | FYA/FYA | 4.28 (2.71–5.42) | 0.592 |
| | FYB/FYB | 4.03 (3.57–4.88) | |
| | FYAB | 4.15 (3.46–5.08) | |

The parasite density is presented as the median (min–max). Statistically significant difference was determined using the Kruskal-Wallis test, and pairwise comparisons were made with Bonferroni correction, as follows: [1] TT vs. CC, [2] TT vs. CT, [3] CT vs. TT, and [4] AG vs. GG.

### 3.4. Genetic Diversity of Pvs25

3.4.1. Polymorphism of Pvs25

The *Pvs25* gene was successfully amplified in 228 *Pv* isolates, including 131 isolates from Sai Yok district (2006–2007) and 97 isolates from Mae Sot district (2014–2016). Three-point mutations (E97Q, I130T, and Q131K) were found in the *Pvs25* gene compared to the reference Sal-I sequence. The *Pvs25* domains, including the signal sequence (SS), and four epidermal growth factor (EGF)-like domains were assigned during the alignment process (Figure 1). The SS, EGF1, and EGF4 domains remained conserved, while the EGF3 domain showed more significant variability with two amino acid substitutions (I130T and Q131K), and EGF2 contained only one point mutation (E97Q). Notably, the prevalence of I130T was 100% (Table 3). Statistically significant differences were observed only for the E97Q mutation when comparing the two sample collection periods ($p = 0.005$). This temporal decrease could be attributed to sample size differences between the two periods or the sample sites.

**Table 3.** The distribution of *Pvs25* polymorphisms in the first (2006–2007) and the second (2014–2016) periods.

| Position | AA Changes | Total *n* (%) | First Period N (%) | Second Period N (%) |
|---|---|---|---|---|
| E97Q [1] | E | 207 (90.8%) | 113 (49.6%) | 94 (42.5%) |
| | Q | 21 (9.2%) | 18 (7.9%) | 3 (1.3%) |
| I130T | T | 228 (100%) | 131 (57.5%) | 97 (42.5%) |
| Q131K | Q | 212 (93%) | 123 (54.0%) | 89 (39%) |
| | K | 16 (7%) | 8 (3.5%) | 8 (3.5%) |

AA, amino acid. [1] Significant difference between first and second period ($p = 0.012$).

3.4.2. Pvs25 Haplotypes

Nonsynonymous amino acid substitutions in *Pvs25* were grouped into four haplotypes (Figure 2). The predominant haplotype, H1(EGTKV), was observed in both areas, accounting for 86.2% and 90.8% of the haplotypes, respectively. This indicates the widespread occurrence of H1 in both locations. H2 (QGTQV) and H3 (QGTKV) also exhibited varying frequencies at the two study sites.

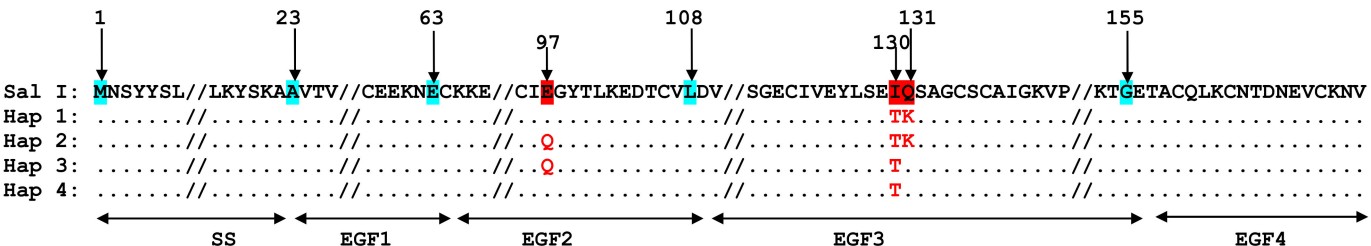

**Figure 2.** Multiple sequence alignments of the four *Pvs25* haplotypes in 228 *Pv* isolates. The sequences were compared with the Salvador I (Sal-I) reference sequence (AF083502), which represents the secretory signal sequence (SS) and the four epidermal growth factor-like domains (EGF-1 to EGF-4). "//" indicates conserved sequences, and dots represent identical amino acid residues. The red color denotes amino acid substitutions identified in the current study, while Cyan color highlights mark the beginning of the EGF domains.

3.4.3. Nucleotide Diversity and Natural Selection of the Pvs25 Gene

The genetic parameters of *Pvs25* were analyzed, revealing varying levels of variation in different regions. The EGF1 and EGF4 domains showed no genetic variation. In the EGF2 region, an average of 0.161 nucleotide differences was observed, with an Hd of $0.161 \pm 0.031$ (mean $\pm$ SD). Similarly, in the EGF3 region, an average of 0.123 nucleotide differences were found, with Hd values of $0.123 \pm 0.029$. Tajima's D-value for both domains

indicated no significant deviation from neutrality ($p > 0.1$). An average of 0.284 nucleotide differences, two segregating sites, and four haplotypes were observed for the full-length gene. A Tajima's D-value of −0.1964 also supports the absence of a significant deviation from neutrality ($p > 0.1$) (Table 4). Significant differences in *Pvs25* gene sequences were found in the isolates collected from both areas. The isolates from Sai Yok displayed greater average numbers of nucleotide differences (K = 0.329) than those from Mae Sot (K = 0.213), indicating greater genetic diversity in the first area.

**Table 4.** Nucleotide diversity and tests of neutrality of the *Pvs25* gene.

|  | K | S | No. Mutations | H | Hd ± SD | $\pi$ ± SD | Tajima's D (*p*-Value) |
|---|---|---|---|---|---|---|---|
| EGF1 | 0 | 0 | 0 | 0 | 0 | 0 | 0 |
| EGF2 | 0.161 | 1 | 1 | 2 | 0.161 ± 0.031 | 0.0012 ± 0.0002 | −0.034 (>0.1) |
| EGF3 | 0.123 | 1 | 1 | 2 | 0.123 ± 0.029 | 0.0009 ± 0.0002 | −0.256 (>0.1) |
| EGF4 | 0 | 0 | 0 | 0 | 0 | 0 | 0 |
| Full length | 0.284 | 2 | 2 | 4 | 0.226 ± 0.036 | 0.0006 ± 0.0001 | −0.1964 (>0.1) |

K, average number of nucleotide differences; S, number of segregating sites; H, number of haplotypes; Hd, haplotype diversity; $\pi$, average pairwise nucleotide diversity; SD, standard deviation.

### 3.4.4. Comparison of Amino Acid Polymorphisms in Thai Pvs25 and the Global Pvs25

A comparative analysis of amino acid polymorphisms in *Pvs25* was performed using publicly available global sequences from 15 countries, comprising 941 *Pvs25* sequences (Figure 3). Compared to the Sal-I sequence, the global *Pvs25* sequence showed a greater level of polymorphism, with 40 haplotypes across the gene. Notably, most of these haplotypes (34 of 44, 77.3%) were single-copy *Pvs25* haplotypes, contributing substantially to the overall haplotype diversity. Most singleton *Pvs25* haplotypes were directly associated with one of the three following major haplotypes: H1, H5, or H6. Among the global *Pvs25* haplotypes, only seven were found at higher frequencies (H1, H2, H5, H6, H7, H8, and H14), indicating the relative conservation of *Pvs25* within the global *Pv* population. Notably, H6 emerged as the most prevalent haplotype (45.2%), exclusive to Asian *Pvs25* from countries such as Bangladesh [27], India [25], China [21,22], Myanmar [20], and Thailand [19]. H2 (23.3%), H6 (45.2%), and H7 (3.7%) were also exclusive to Asian *Pvs25*, while H5 and H8, with frequencies of 17.4% and 2.6%, respectively, were shared by the Asian and Mexican *Pvs25* populations. H1 and H14 originated in Latin America (Mexico [28], Venezuela, and Brazil [27]) and were directly linked to each other. The amino acid variants of the *Pvs25* protein are succinctly summarized in Table 5. Among the 32 global variants of the *Pvs25* sequence, the most common amino acid substitution was I130T, which was detected in all Asian isolates and Mexican [28], Brazilian, Mauritanian, and Papuan isolates [27]. The Q131K variant was present in certain Asian isolates from Thailand, Myanmar, India, and Bangladesh, as well as in PNG isolates. The Q87K variant was identified in North America (Mexico), South America (Brazil, Colombia, and Venezuela), and Africa (Mauritania). Notably, the Q87K and Q87L substitutions were observed in isolates from Iran and China, respectively. Additionally, the E97Q amino acid residue was identified in several Asian isolates but was absent in American isolates. Other nonsynonymous mutations have been reported in *Pvs25* at very low frequencies. Importantly, no sequence identical to the Sal-I sequence was found within the global *Pvs25* population.

**Table 5.** Global comparisons of amino acid polymorphisms in the *Pvs25* sequences.

| Geographic origin | N | SS | | EGF1 | | | | | | | EGF2 | | | | | | | | | | | | | | | | | EGF4 | | | |
|---|---|---|---|---|---|---|---|---|---|---|---|---|---|---|---|---|---|---|---|---|---|---|---|---|---|---|---|---|---|---|---|
| | | 2 * | 27 | 35 | 38 | 40 | 45 | 55 | 61 | 63 | 79 | 86 | 87 | 97 | 101 | 106 | 123 | 130 | 131 | 132 | 135 | 136 | 146 | 153 | 156 | 158 | 159 | 161 | 162 | 178 | 180 |
| Sal-I strain | | N | D | L | M | N | M | E | K | E | C | A | Q | E | L | C | I | I | Q | S | C | S | E | K | E | A | C | L | K | K | Q |
| India | 100 | - | - | - | - | - | - | - | - | - | C/R | - | - | E/Q | - | - | - | T | Q/K | - | - | - | - | - | - | - | - | - | - | - | - |
| Iran | 4 | - | - | - | - | - | - | - | - | - | - | - | Q/K | E/Q | - | - | - | T | - | - | - | - | - | - | - | - | - | - | - | - | - |
| China | 354 | - | D/N | L/M | - | - | - | - | - | - | - | - | Q/L | E/Q | - | - | - | T | Q/K | - | - | - | - | - | - | - | - | - | - | - | - |
| Thailand | 3 | - | - | - | - | - | - | - | - | - | - | - | - | E/Q | - | - | - | T | Q/K | - | - | - | - | - | - | - | - | - | - | - | - |
| Myanmar | 62 | - | - | - | - | N/S | - | E/G | - | - | - | - | - | E/Q | - | - | - | T | Q/K | - | - | S/P | E/G | - | - | A/V | - | - | - | - | - |
| South Korea | 100 | N/D | - | L/P | - | - | M/T | - | K/C | E/K | - | A/P | - | E/Q | L/S | C/R | I/F | T | - | - | C/G | - | - | K/R | E/D | - | C/R | L/W | K/R | K/R | Q/E |
| North Korea | 1 | - | - | - | - | - | - | - | - | - | - | - | - | - | - | - | - | T | - | - | - | - | - | - | - | - | - | - | - | - | - |
| Bangladesh | 4 | - | - | - | - | - | - | - | - | - | - | - | - | E/K | - | - | - | T | Q/K | - | - | - | - | - | - | - | - | - | - | - | - |
| Indonesia | 1 | - | - | - | - | - | - | - | - | - | - | - | - | Q | - | - | - | T | - | - | - | - | - | - | - | - | - | - | - | - | - |
| Mexico | 64 | - | - | - | - | - | - | - | - | - | - | - | Q/K | - | - | - | - | I/T | - | - | - | - | - | - | - | - | - | - | - | - | - |
| Colombia | 1 | - | - | - | - | - | - | - | - | - | - | - | K | - | - | - | - | - | - | - | - | - | - | - | - | - | - | - | - | - | - |
| Venzuela | 16 | - | - | - | M/T | - | - | - | - | - | - | - | Q/K | - | - | - | - | - | - | - | C/R | S/R | - | - | - | - | - | - | - | - | - |
| Brazil | 1 | - | - | - | - | - | - | - | - | - | - | - | Q/K | - | - | - | - | I/T | - | - | - | - | - | - | - | - | - | - | - | - | - |
| PNG | 1 | - | - | - | - | - | - | - | - | - | - | - | - | - | - | - | - | T | K | R | - | - | - | - | - | - | - | - | - | - | - |
| Mauratinea | 1 | - | - | - | - | - | - | - | - | - | - | - | K | - | - | - | - | T | - | - | - | - | - | - | - | - | - | - | - | - | - |
| Present Study | 228 | - | - | - | - | - | - | - | - | - | - | - | - | E/Q | - | - | - | T | Q/K | - | - | - | - | - | - | - | - | - | - | - | - |
| **Overall** = 16 countries | | | | | | | | | | | | | | | | | | | | | | | | | | | | | | | |
| N of substitutioins | | 1 | 1 | 2 | 1 | 1 | 1 | 1 | 1 | 1 | 1 | 1 | 7 | 9 | 1 | 1 | 1 | 14 | 7 | 1 | 2 | 2 | 1 | 1 | 1 | 1 | 1 | 1 | 1 | 1 | 1 |
| % Frequency | | 6.3 | 6.3 | 12.5 | 6.3 | 6.3 | 6.3 | 6.3 | 6.3 | 6.3 | 6.3 | 6.3 | 43.8 | ### | 6.3 | 6.3 | 6.3 | 87.5 | 43.8 | 6.3 | 12.5 | 12.5 | 6.3 | 6.3 | 6.3 | 6.3 | 6.3 | 6.3 | 6.3 | 6.3 | 6.3 |

*SS*, secretary signal sequence, EPF, epidermal growth factor, * indicates identical amino acid residues compared to the Sal-I strain.

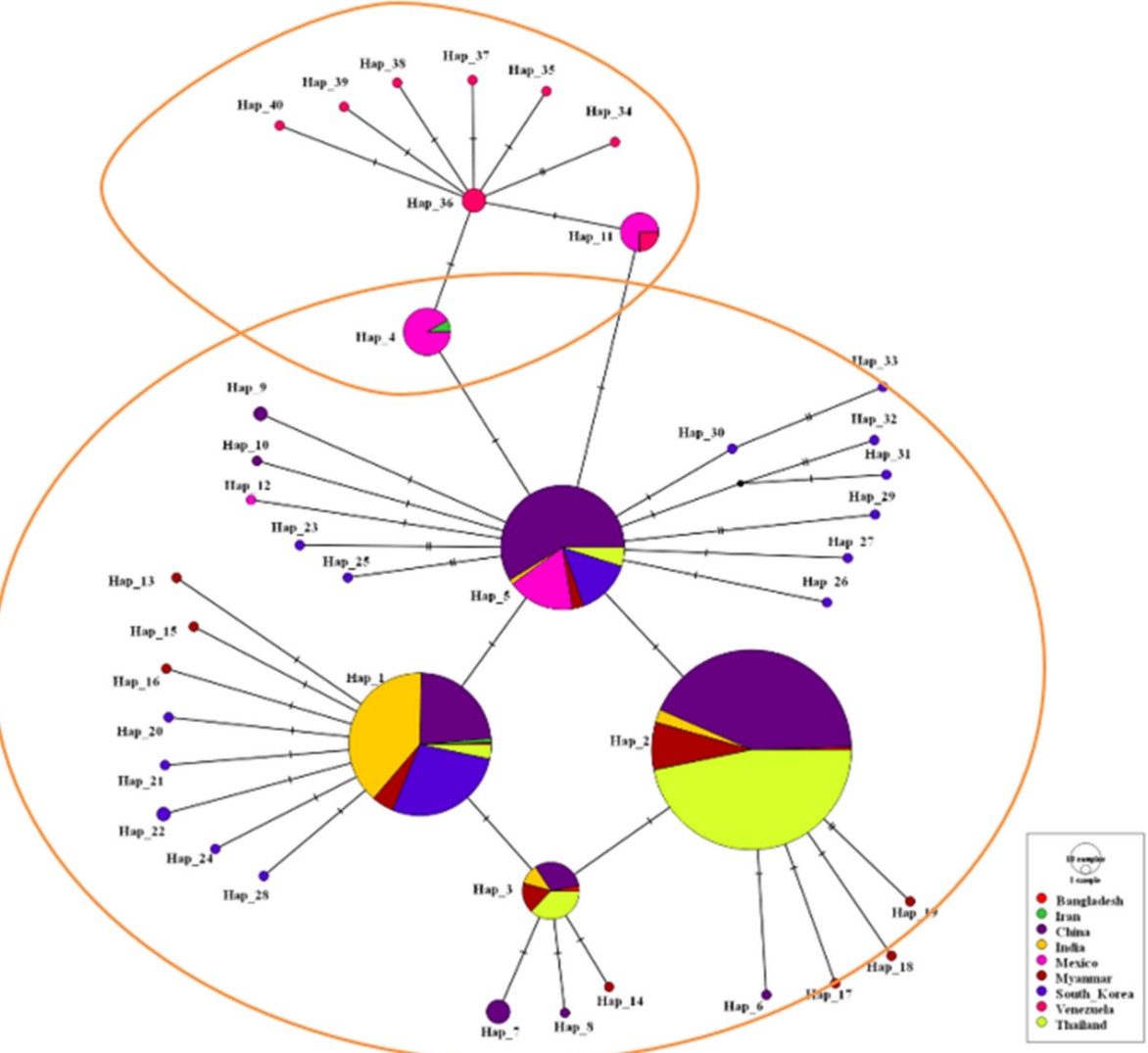

**Figure 3.** Haplotype network analysis of the global *Pvs25* dataset. The number of branches is proportional to the divergence. The size of each circle indicates the proportion of the total haplotype frequency. The color of each node corresponds to a different geographic origin.

## 4. Discussion

### 4.1. Genetic Diversity of Host Genes

#### 4.1.1. TLRs and Adaptor Proteins

Understanding host genetic factors linked to *Pv*-malaria susceptibility or resistance is crucial for effective control, elimination strategies, and vaccine development [14]. Innate immune responses, mediated by PRRs, like TLRs, RIG-I-like receptors (RLRs), and scavenger receptors, contribute significantly to infectious disease progression [29,30].

In particular, TLRs bridge innate and adaptive immunity by activating essential transcription factors and inducing pro-inflammatory cytokines and chemokines [31]. Genetic variations in TLR polymorphisms significantly affect disease susceptibility and clinical outcomes [32]. The TLR6, TLR9, TIRAP, and TOLLIP variants are involved in the TLR signaling pathway, which plays a crucial role in the immune response. Activation of this pathway involves the adaptor protein (TIRAP), which mediates downstream signaling and triggers pro-inflammatory cytokines. These cytokines aid the host in eliminating parasites [33]; however, an excessive increase in these cytokines can lead to systemic inflammation and severe disease development [34].

In this study, specific genetic variants, such as TLR9 rs187084 T, TOLLIP rs3750920 C/T, and ICAM-1 rs5498 A/G, were associated with increased parasite load. Deviations in genotype frequencies from HWE were observed for TIRAP, TLR6, TLR9, TOLLIP rs5743899, and DARC, suggesting potential factors like gene mutations, genetic drift, nonrandom mating, population migration, or natural selection [35].

The TLR6 rs5743810 promoter polymorphism was the most common homozygous wild type (TT genotype), consistent with the findings of previous studies [15,36], in which the TT genotype was the most prevalent, followed by the CT and CC genotypes. However, the study population lacked different clinical manifestations of *Pv* malaria (asymptomatic, mild, and severe), preventing intragroup genotype comparisons. The TLR9 rs187084 C/T genotype was the most frequent, while the heterozygous genotype was not found for TLR9 rs5743836. In previous studies, the TLR9 rs187084 C/T polymorphism has been associated with susceptibility to *Pv* malaria and placental malaria caused by *Pf* [12,37]. In contrast, studies in Brazil [12] and Iran [38] have shown no evidence to support the influence of TLR9 promoter polymorphisms on susceptibility to mild malaria in the studied populations. The observed genotype and allele frequencies of TLR9 rs187084 in our study population were consistent with those of a study in the United States, which reported high heterozygosity across the following three ethnic groups: Hispanic Americans, European Americans, and African Americans [39]. Interestingly, studies in India [40] and Brazil [15] reported contrasting findings, possibly attributable to differences in ethnic populations with diverse genetic backgrounds or varying endemic contexts.

Regarding the TIRAP S180L (rs8177374) variant, our study found that the CC genotype was the most common and was previously considered a risk factor for severe *Pv*-malaria infection due to an uncontrolled immune response [41]. This study also found a low occurrence of heterozygotes (CT genotype). However, a study conducted in Vietnam, Kenya, and Gambia found that TIRAP S180L C/T had a significant protective effect against mild and severe malaria. In contrast, a study in a Brazilian population did not find any association between this genotype and mild malaria [12]. An Iranian study revealed that individuals with the TIRAP S180L heterozygote have a 1.5-fold increased risk of mild malaria [38]. This protective effect may be due to the ability of this gene to attenuate TLR2 and TLR4 signal transduction, resulting in reduced production of inflammatory cytokines [38,42]. Homozygote mutants were found in 4.0% of the study population, consistent with the findings of a previous study reporting a low occurrence of this genotype in the United Kingdom [42]. However, homozygous mutants were not found in Kenyan or West African populations of Guinea-Bissau, Gambia, or the Republic of Equatorial Guinea. The absence of this genotype in African populations could be attributed to strong selection pressures against homozygous mutant individuals, which reduces the frequency of the Leucine180 allele [42].

The TOLLIP adaptor protein, which is crucial for regulating the TLR signaling pathway, exhibited genetic polymorphisms associated with various infections, such as malaria, tuberculosis, and leishmaniasis [14,43,44]. In our study, heterozygotes of AG and CT were dominant, with similar frequencies of heterozygous genotypes (AG, 41%; CT, 42%) reported in Brazilian individuals infected with *Pv* [14]. In addition, G and T alleles frequencies accounted for 60.4% and 33.5%, respectively. In the Brazilian population, both alleles were linked to a higher risk of cutaneous leishmaniasis [43]. In contrast, these alleles were found to have a protective effect against lepromatous leprosy in the Mexican population [45]. The rs3750920 T allele is a risk factor for developing malaria and cutaneous leishmaniasis [14,43]. The allele frequency of rs3750920 T observed in this population (60%) was greater than that reported in Vietnamese (35%), African (33%), and Caucasian (47%) populations [44]. Genotype or allele frequencies vary across populations because of genetic variations among different ethnic groups, arising from genetic drift, migration patterns, natural selection, or genetic admixture [35]. Therefore, the frequency disparities in genetic variants likely result from differences among various ethnic groups.

Association analysis between TLR6 rs5743810 and parasite density showed no statistically significant differences among the genotypes. Similarly, TLR9 rs5743836 was not significantly associated with parasite density. It is worth noting that an association between the TT homozygous genotype of this variant and low parasite load was observed in the Amazonian region of Brazil [12]. The TLR9 rs187084 T genotype showed varied correlations with parasite density, contrasting with findings in other populations [15]. TOLLIP gene analysis suggested that the CT genotype significantly contributed to increased susceptibility to *Pv* infection. However, associations between host genotypes and parasite load in Brazilian patients [14] did not align with our study, potentially due to geographical and genetic differences.

### 4.1.2. Parasite-Binding Genes

The increasing occurrence of severe cases of *Pv* infection suggests shared pathogenic mechanisms with *Pf* malaria, such as cytoadherence. In *Pf*, cytoadhesion to the human endothelium involves interactions between members of the *Pf.* erythrocyte membrane protein 1 (*PfEMP-1*) family and host receptors on endothelial cells such as CD36 and ICAM-1 [46,47]. The adhesion of *PfEMP1* to ICAM-1 is associated with severe *Pf* malaria and has been implicated in the cytoadhesion of *Pv* parasites [47]. The frequencies of ICAM-1 genotypes showed that homozygous wild type was the most prevalent, whereas the homozygote mutant (GG genotype) was the least frequent (6%). Similar genotype frequencies of this gene were reported in an earlier study in Thailand [35], although an Indian study reported a different genotype distribution [48]. Such variations in genotype distribution can be attributed to genetic differences among populations. Regarding parasite load, the heterozygous AG carriers exhibited higher parasite density, contrasting with findings in Thai patients infected with *Pf* [35].

*Pv* also relies on the DARC or Duffy antigen (Fy) for reticulocyte entry, with variations in $Fy^a$ and $Fy^b$ antigens on the red cell surface affecting the host's susceptibility to *Pv* malaria [49]. The most common genotype was FYA/FYA, followed by the heterozygous FYA/FYB genotype. Furthermore, the association of the FY genotype with parasitemia revealed no statistically significant correlation, underscoring the complexity and inconclusive nature of host genetic polymorphism studies in malaria susceptibility.

### 4.2. Genetic Diversity of Pv

Understanding *Pv* transmission dynamics and population genetics is crucial for effective malaria control, especially in addressing drug resistance and predicting parasite phenotypes in malaria-endemic areas [50]. This knowledge is vital for implementing control strategies and developing malaria vaccines [51]. However, the remarkable ability of *Pv* to generate numerous polymorphisms, potentially leading to immune evasion, complicates the design of a universally effective vaccine [26]. To address this challenge, identifying and utilizing relatively conserved *Pv* genes are crucial for vaccine development. The patients from Mae Sot district (Tak province) and Sai Yok district (Kanchanaburi province) represent mostly full-time residents, transient or migrant populations. Several factors contributing to genetic diversity were not investigated in the current study. Although all participants were of Thai descent, it is worth noting that some may cross into Myanmar because of the area's proximity to the border. This study focused on *Pvs25*, a promising transmission-blocking vaccine (TBV) candidate [22], that comprises three distinct sections, namely, an initial N-terminal signal peptide sequence, four EGF-like domains, and a glycosylphosphatidylinositol (GPI) anchor [52]. *Pvs25* exhibited nucleotide polymorphisms at three sites (E97Q, I130T, and Q131K), with the most frequent mutations observed in the EGF3 domain, which is recognized as a blocking antibody epitope in *Pvs25* [53]. The observed amino acid changes in our isolates were in line with those of a previous report from Thailand [19], indicating low genetic diversity in the *Pvs25* gene. The limited diversity of the sexual stage antigen may be linked to the expression of its protein (P25) during mosquito stages. This enables it to evade immune selection within the human host, possibly contributing to the

long-term conservation of *Pvs25* [22]. Moreover, the E97Q mutant point only showed a statistically significant difference when comparing the two sample collection periods. This temporal change could be attributed to differences in the sample sizes between the two periods or the study sites. This study also identified four haplotypes of *Pvs25*, consistent with findings from Myanmar and China [20,21]. The nucleotide diversity of *Pvs25* in this study ($\pi$ = 0.0006) aligns with that reported in other Southeast Asian countries [20,21], indicating the relative conservation of the gene in the region. Amino acid changes and haplotype network analysis of *Pvs25* revealed distinct genetic variations in the global *Pv* population. Asian and American/African *Pvs25* genetic variants share some amino acid changes, with significant substitutional differences in these populations. For instance, the E97Q substitution is exclusive to Asian *Pvs25*, while the I130T and Q131K variants are more prevalent in Asian populations. In contrast, the Q87K substitution is more common in American and African populations. Such geographic differences in *Pvs25* may affect vaccine effectiveness in different regions, emphasizing the need for region-specific TBV development. The limitation of the current study is that parasite diversity isolated from mosquitoes was not included in the study. It is worth noting that the previous studies examined *Pv*CSP-*Pvs25* [54] or *Pvs25*-*Pvs28* [28] haplotypes from field isolates for infectivity to various Anophelines mosquito species. However, our study explored only the genetic variants for the *Pvs25* gene. Therefore, correlating our *Pvs25* finding with the susceptibility of Anopheline mosquito species, as studied previously, is not possible.

## 5. Conclusions

This study evaluated the genetic diversity of both the parasite and the host. In *Pvs25* gene, three amino acid changes were identified, with the EGF3 domain exhibiting the most frequent alteration. Despite the overall genetic conservation of *Pvs25* in Thai *Pv* isolates and Southeast Asian countries, notable genetic variations were observed between Asian and American/African populations, suggesting substantial geographic differentiation. Moreover, the study explored genetic variations in innate immune response genes, particularly TLRs and adaptor molecules, such as TIRAP and TOLLIP, which contribute to controlling parasitemia and influencing disease manifestations in the host. The associations between TLR9 rs187084 T, TOLLIP rs3750920 C/T, and ICAM -rs5498 A/G and high parasite density in patients infected with *Pv* suggest their potentially critical role in malaria susceptibility. The associations between specific genetic variants (such as TLR9 rs187084 T, TOLLIP rs3750920 C/T, and ICAM rs5498 A/G) and increased parasite load suggest their potential influence on the treatment outcomes. Deviations from the HWE in specific studied SNPs point to factors such as gene mutations, genetic drift, population migration, or natural selection influencing these genetic patterns. Future studies should investigate the impact of these genetic variations on parasite fitness, the immune response, and transmission dynamics in various geographic contexts.

**Author Contributions:** Conceptualization, K.N.-B. and W.C.; methodology, W.C. and A.A.J.; validation, W.C. and A.A.J.; sample collection, K.N.-B. and W.C.; data analysis, W.C. and K.N.-B.; original draft preparation, A.A.J.; manuscript finalization, K.N.-B.; funding acquisition, K.N.-B. All authors have read and agreed to the published version of the manuscript.

**Funding:** This research project was supported by the Thailand Science Research and Innovation Fundamental Fund Fiscal Year 2024 and Thammasat University (Chulabhorn International College of Medicine, Center of Excellence in Pharmacology and Molecular Biology of Malaria and Cholangiocarcinoma). Kesara Na-Bangchang is supported by the National Research Council of Thailand under a Research Team Promotion grant.

**Institutional Review Board Statement:** The study protocol was approved by the Ethics Committee of Thammasat University (No. 082/2560).

**Informed Consent Statement:** All participants provided written informed consent for participation in the study.

**Data Availability Statement:** Data will be made available upon request.

**Acknowledgments:** We are thankful to the staff of the malaria clinics or hospitals in Sai Yok (Kanchanaburi province) and Mae Sot (Tak province) for their kind assistance during sample collection.

**Conflicts of Interest:** The authors declare no conflicts of interest. The funders had no role in the study design, collection, analysis, interpretation of the data, writing of the manuscript, or decision to publish the results.

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
