# Peer review of "Genetic Diversity of Plasmodium vivax Surface Ookinete Protein Pvs25 and Host Genes in Individuals Living along the Thai–Myanmar Border and Their Relationships with Parasite Density"

_2036-7481, doi:10.3390/microbiolres15020045_

Round 1

Reviewer 1 Report

Comments and Suggestions for Authors

Jalei et al. investigated genetic diversity in the Plasmodium vivax gene encoding the ookinete surface protein, pv25. Polymorphisms in the human genes associated with pattern recognition in innate immunity and adhesion were also investigated. The study makes important contributions that will help with the continued efforts toward vaccine development for P. vivax.

Comments:

1.      Include a map of the Thai-Myanmar border area.

2.      The population studied needs additional definitions. It is not clear if the patients from Mae Sot district (Tak province) and Sai Yok district (Kanchanaburi province) represent mostly full time residents, transient or migrant populations. This is important in understanding the genetic patterns observed and the conclusions reach by the investigators in lines 442-446 with regards to this particular area of the Thai-Myanmar border. Conclusions should be cautious as many factors contributing to genetic diversity were not investigated in the current study.

3.      There is no indication of the allele distribution of the rs3750920T and TLR9 rs187084T alleles in Myanmar as susceptibility and high parasite densities are associated with both alleles.

4.      The mosquito vectors transmitting P. vivax in the area of the study and neighboring neighboring countries should be included. Is the genetic variation in the Pv25 gene consistent with data obtained from blood collected from patients and from parasites obtained from vectors?

5.      Line 215: Amino acid  

Comments on the Quality of English Language

Minor English corrections.

Author Response

Reviewer 1

Jalei et al. investigated genetic diversity in the Plasmodium vivax gene encoding the ookinete surface protein, pv25. Polymorphisms in the human genes associated with pattern recognition in innate immunity and adhesion were also investigated. The study makes important contributions that will help with the continued efforts toward vaccine development for P. vivax.

Comments:

  1. Include a map of the Thai-Myanmar border area.

Response: The map of Thailand has been added (Figure 1).

2.The population studied needs additional definitions. It is not clear if the patients from Mae Sot district (Tak province) and Sai Yok district (Kanchanaburi province) represent mostly full-time residents, transient or migrant populations. This is important in understanding the genetic patterns observed and the conclusions reached by the investigators in lines 442-446 with regard to this particular area of the Thai-Myanmar border. Conclusions should be cautious as many factors contributing to genetic diversity were not investigated in the current study.

Response:

Thank you very much for this valuable comment. Although all participants were of Thai descent, it is worth noting that some may cross into Myanmar due to the proximity of the area to the border. This has been added in the discussion.

  1. There is no indication of the allele distribution of the rs3750920T and TLR9 rs187084T alleles in Myanmar as susceptibility and high parasite densities are associated with both alleles.

Response:

There is no previous report of the rs3750920T and TLR9 rs187084T alleles in Myanmar. The available data on rs3750920T and TLR9 rs187084T alleles and parasite density were reported by Omar et al., 2012, from Ghanaian children, which showed a significant association of TLR9 gene polymorphisms with symptomatic malaria (P. falciparum) among Ghanaian children.

  1. The mosquito vectors transmitting P. vivax in the area of the study and neighboring countries should be included. Is the genetic variation in the Pv25 gene consistent with data obtained from blood collected from patients and from parasites obtained from vectors?

Response:

This is the limitation of the current study; it did not include the mosquito samples. However, there are some reports of parasites from mosquitoes containing a similar pattern of Pvs25 as found in human (González-Cerón et al., 2019).

Reference: González-Cerón L, Rodríguez MH, Nettel-Cruz JA, Hernández-Ávila JE,Malo-García IR, Santillán-Valenzuela F, Villarreal-Treviño C (2019). Plasmodium vivax CSP-Pvs25 variants from southern Mexico produce distinct patterns of infectivity for Anopheles albimanus versus An. pseudopunctipennis, in each case independent of geographical origin. Parasites and Vectors 12:86.

  1. Line 215: Amino acid  

Response: This has been corrected as suggested.

Reviewer 2 Report

Comments and Suggestions for Authors

The study utilizes a classical genetic method PCR with RFLP to address a complex aspect of malaria research, focusing on the genetic diversity of Plasmodium vivax and its interaction with host genes. By focusing on the Thai-Myanmar border, the study adds valuable regional insights into malaria epidemiology and genetic diversity. This regional focus is essential for tailoring public health interventions and understanding the dynamics of malaria transmission in specific contexts. This comprehensive approach is crucial for developing more effective malaria control strategies and understanding the mechanisms of disease susceptibility and resistance. This manuscript is well-written and informative. There are some minor comments below:

1) It might be beneficial to explicitly state the study's main objective early in the abstract for clarity.

2) In the Methods section, while statistical tools and general approaches are mentioned, more specific details about the statistical models or tests used for certain analyses could enhance the reproducibility of the study.  

3) In figure 1 legend,  what’s the definition of blue or red color for highlighting?

4) The discussion could be strengthened by deeper analysis and comparison with existing studies. Highlighting how the findings align with or differ from other research adds to the paper's contribution and situates it within the broader field.

Author Response

Reviewer 2

The study utilizes a classical genetic method PCR with RFLP to address a complex aspect of malaria research, focusing on the genetic diversity of Plasmodium vivax and its interaction with host genes. By focusing on the Thai-Myanmar border, the study adds valuable regional insights into malaria epidemiology and genetic diversity. This regional focus is essential for tailoring public health interventions and understanding the dynamics of malaria transmission in specific contexts. This comprehensive approach is crucial for developing more effective malaria control strategies and understanding the mechanisms of disease susceptibility and resistance. This manuscript is well-written and informative. There are some minor comments below:

  1. It might be beneficial to explicitly state the study's main objective early in the abstract for clarity.

Response: The aim of this study has been added in the Abstract section.

  1. In the Methods section, while statistical tools and general approaches are mentioned, more specific details about the statistical models or tests used for certain analyses could enhance the reproducibility of the study.  

Response: The statistical method used in the current study has been added as “The Kruskal‒Wallis test with Dunn’s post hoc test or Mann‒Whitney U test was performed to determine the association between genotypes, gene alleles, and Pv parasite density.

  1. In figure 1 legend, what’s the definition of blue or red color for highlighting?

Response: Figure 1 is now changed to Figure 2. Additionally, the legend now includes explanations for the blue and red color text highlights: blue indicates the initiation of domains, while red indicates amino acid substitutions identified in the current study."

  1. The discussion could be strengthened by deeper analysis and comparison with existing studies. Highlighting how the findings align with or differ from other research adds to the paper's contribution and situates it within the broader field.

Response: Most of the previous data showed that host and parasite factors might be impacted by P. falciparum parasite density. However, the current study showed that both factors (host and parasites) are also affected by the P. vivax parasite densities.

Round 2

Reviewer 1 Report

Comments and Suggestions for Authors

To better clarify the data, and since mosquito studies were not conducted, provide a brief explanation for the similarities (and differences) in Pvs25 gene diversity obtained in mosquitoes (Ref. 54) to the data obtained in the current study.

Author Response

Reviewer 1:

To better clarify the data, and since mosquito studies were not conducted, provide a brief explanation for the similarities (and differences) in Pvs25 gene diversity obtained in mosquitoes (Ref. 54) to the data obtained in the current study.

Response

Thank you very much for your comment. It is worth noting that the previous studies examined Plasmodium vivax CSP-Pvs25 [1] or Pvs25-Pvs28 [2] haplotypes from field isolates for infectivity to various Anophelines mosquito species. However, our study explored only the genetic variants for the Pvs25 gene. Therefore, correlating our Pvs25 finding with the susceptibility of Anopheline mosquito species, as studied previously, is not possible. This has been added in the Discussion section.

References

  1. González-Cerón L, Rodríguez MH, Nettel-Cruz JA, Hernández-Ávila JE, Malo-García IR, Santillán-Valenzuela F, et al. Plasmodium vivax CSP-Pvs25 variants from southern Mexico produce distinct patterns of infectivity for Anopheles albimanus versus An. pseudopunctipennis, in each case independent of geographical origin. Parasites & Vectors. 2019; 12:86.
  2. González-Cerón L, Alvarado-Delgado A, Martínez-Barnetche J, Rodríguez MH, Ovilla-Muñoz M, Pérez F, et al. Sequence variation of ookinete surface proteins Pvs25 and Pvs28 of Plasmodium vivax isolates from Southern Mexico and their association to local anophelines infectivity. Infection, Genetics and Evolution. 2010; 10:645-54.